# South Africa's male homicide epidemic hiding in plain sight: Exploring sex differences and patterns in homicide risk in a retrospective descriptive study of postmortem investigations

**Richard Matzopoulos**[1,2]*, **Megan R. Prinsloo**[1,2,3], **Shibe Mhlongo**[4], **Lea Marineau**[5], **Morna Cornell**[6], **Brett Bowman**[7], **Thakadu A. Mamashela**[8], **Nomonde Gwebushe**[9], **Asiphe Ketelo**[4], **Lorna J. Martin**[10], **Bianca Dekel**[4], **Carl Lombard**[9,11], **Rachel Jewkes**[4,12], **Naeemah Abrahams**[4,13]

1 Burden of Disease Research Unit, South African Medical Research Council, Cape Town, South Africa, 2 Division of Public Health Medicine, School of Public Health, Faculty of Health Sciences, University of Cape Town, Cape Town, South Africa, 3 Institute for Lifecourse Development, Faculty of Education, Health & Human Sciences, University of Greenwich, London, United Kingdom, 4 Gender and Health Research Unit, South African Medical Research Council, Cape Town, South Africa, 5 Johns Hopkins University School of Nursing, Baltimore, MD, United States of America, 6 Centre for Infectious Disease Epidemiology & Research, School of Public Health, Faculty of Health Sciences, University of Cape Town, Cape Town, South Africa, 7 School of Human and Community Development, University of the Witwatersrand, Johannesburg, South Africa, 8 Department of Forensic Medicine, Faculty of Health Sciences, University of Limpopo, Polokwane, South Africa, 9 Biostatistics Unit, South African Medical Research Council, Cape Town, South Africa, 10 Division of Forensic Medicine and Toxicology, Faculty of Health Sciences, University of Cape Town, Cape Town, South Africa, 11 Division of Epidemiology and Biostatistics, Department of Global Health. Stellenbosch University, Cape Town, South Africa, 12 Office of the Executive Scientist, South African Medical Research Council, Cape Town, South Africa, 13 Division of Social and Behavioural Sciences, School of Public Health, Faculty of Health Sciences, University of Cape Town, Cape Town, South Africa

* richard.matzopoulos@uct.ac.za

**Data Availability Statement:** Availability of data used in the study would be subject to permission

## Abstract

South Africa has an overall homicide rate six times the global average. Males are predominantly the victims and perpetrators, but little is known about the male victims. For the country's first ever study on male homicide we compared 2017 male and female victim profiles for selected covariates, against global average and previous estimates for 2009. We conducted a retrospective descriptive study of routine data collected through postmortem investigations, calculating age-standardised mortality rates for manner of death by age, sex and province and male-to-female incidence rate ratios with 95% confidence intervals. We then used generalised linear models and linear regression models to assess the association between sex and victim characteristics including age and mechanism of injury (guns, sharp and blunt force) within and between years. 87% of 19,477 homicides in 2017 were males, equating to seven male deaths for every female, with sharp force and firearm discharge being the most common cause of death. Rates were higher among males than females at all ages, and up to eight times higher for the age group 15–44 years. Provincial rates varied overall and by sex, with the highest comparative risk for men vs. women in the Western Cape Province (11.4 males for every 1 female). Male homicides peaked during December

by the Health Research Ethics Committee and provincial authorities that approved the original study. This is a recently completed study and the dataset will initially be used for capacity development among the emerging researchers on study team. Thereafter access to a de-identified dataset is available upon reasonable request. Requests should be sent to the convenor of the South African Medical Research Council's Research Ethics Office, Ms Adri Labuschagne (Adri.Labuschagne@mrc.ac.za), for consideration. Guidelines for applications and related materials are available at: https://www.samrc.ac.za/research/rio-research-ethics-office A period of 24 months after publication of the main study results should elapse before requests are made, to allow the authors to publish sub-studies and further analyses.

**Funding:** The study was funded by the South African Medical Research Council (42060 to RM) and the Ford Foundation (133505 to NA). RM, MP, SM, NG, AK, BD, CL, RJ and NA receive salaries from the SAMRC. This work was supported by the National Institutes of Health (U01AI069924 & R01 MH122308-01A1 to MC) and the Fogarty International Center and the National Institute of Mental Health (D43TW011308 to MC). The funders had no role in study design, data collection and analysis, decision to publish, or preparation of the manuscript.

**Competing interests:** I have read the journal's policy and the authors of this manuscript have the following competing interests: BB serves on the board of non-governmental organisation, Gun Free South Africa, but receives no remuneration for this role. The authors declare that the research was conducted in the absence of any commercial or financial relationships that could be construed as a potential conflict of interest.

and were highest during weekends, underscoring the prominent role of alcohol as a risk factor. There is a massive, disproportionate and enduring homicide risk among South African men which highlights their relative neglect in the country's prevention and policy responses. Only through challenging the normative perception of male invulnerability do we begin to address the enormous burden of violence impacting men. There is an urgent need to address the insidious effect of such societal norms alongside implementing structural interventions to overcome the root causes of poverty, inequality and better control alcohol and firearms.

## Introduction

In South Africa, injury-related mortality accounted for 8.6% of deaths in 2009 [1] primarily due to extremely high homicide rates, which were nearly six times the global average [2, 3]. Adult men, age 20 years and older, accounted for more than three-quarters (79%) of all homicides in which the age of the decedent was known [2]. Despite this significant difference, there has been limited focus on the male victims of homicide. Previously, two nationally conducted homicide studies in 1999 and 2009 explored the situational contexts of homicide, but only for women and children victims [4, 5]. This is consistent with global directives such as the 67th World Health Assembly Resolution that have prioritised preventive efforts to reduce violence against women regardless of the higher prevalence and proportion of victims being male. Globally men bear a far higher injury mortality and morbidity burden than women, [6] yet we were unable to identify any studies that explored the different patterns of male and female homicide in South Africa.

To address this gap, the South African Medical Research Council (SAMRC) funded a comprehensive Female and Male Homicide and Injury Mortality Study (FAMHIS) for 2017. The first phase included a nationally representative all-cause injury mortality study. A second phase was specifically designed to collect more detailed information from interviews with police investigating officers. This replicates female homicide studies conducted in 1999 and 2009, [4, 5] and, for the first time, provides comparable information describing the personal and situational risks for male victims.

The objective of our study was to compare (1) male and female victim profiles by external cause, age, province, day of week, month and alcohol-relatedness and (2) male: female homicide rate ratios against global averages for selected covariates (external cause and age), and (3) explored whether the odds of male versus female homicide by external cause and age had changed between 2009 and 2017.

## Methods

### Study design and data sources

We conducted a retrospective descriptive study of routine postmortem investigation data via a nationally representative survey of mortuaries sampled from eight of South Africa's nine provinces for all deaths in 2017. Data were obtained from postmortem reports and ancillary documentation, including police reports and hospital records. For the ninth province, the Western Cape Province, the survey data were combined with compatible routinely captured data from the provincial Forensic Pathology Service (FPS), which maintains these data for all 16 medico-legal mortuaries in the province.

## Sampling

We drew a multistage stratified cluster sample for eight provinces, using mortuaries as the primary sampling unit (cluster). We used a sampling frame of 58,641 postmortem reports from 121 mortuaries to draw a representative sample stratified by province and mortuary size: small (≤500 cases), medium (501–1500 cases) and large (>1500 cases). Sixty-five mortuaries from eight provinces were selected with an expected sample of 22,733 records. Fieldwork was conducted from 20 January to 3 July 2020. To account for the selection probabilities of mortuaries within survey strata, we applied analysis weights. In total 22,822 deaths were included in the survey, which exceeded the expected sample by 89 cases. For the ninth province (Western Cape Province) we appended 8174 records obtained from the provincial FPS. After application of sample weights total deaths due to injury were estimated at 54,734. Further details on sampling, fieldwork and data collection methods are provided elsewhere [7].

## Case selection and variables

Information collected from the postmortem report included age and sex of the deceased and date, external cause and apparent manner of death and blood alcohol concentration. Sex was inferred from the biological sex recorded in the postmortem report. Blood alcohol was analysed using gas chromatography with flame ionization detection method. We excluded all deaths from natural causes, foetal deaths and deaths that occurred outside South Africa. For deaths due to external causes, we excluded suicide and deaths that were transport-related or unintentional after redistributing deaths due to undetermined intent. For all homicides we ascribed an external cause of death consistent with the tenth revision of the International Statistical Classification of Diseases and Related Health Problems, 2007 (ICD-10; S1 Table). We defined weekends as Saturdays and Sundays and hot months as November through March. The mortuary death register number and death notification number were collected as identifiers for follow-up, and to resolve data capture errors, but were excluded from the analysis.

## Statistical analysis

We calculated age-standardised homicide rates (ASHR) by sex using 2017 population estimates provided by Dorrington (2013) [8] and the World Health Organization's (WHO's) world standard population, and male-to-female incidence rate ratios (IRRs) with 95% confidence intervals (CIs). Cases with unknown age were proportionally redistributed for males and females as follows:

$$ASHRr = s \cdot ASHRe$$

$$and \ s = \frac{t}{t - u}$$

where ASHRr = ASHR with unknown age cases redistributed;
ASHRe = ASHR with unknown age cases excluded;
s = scaling factor; t = total number of homicides; u = homicides with unknown age

Similarly, we applied scaling factors to calculate adjusted homicide rates for comparison with the Global Burden of Disease study, which redistributes injury deaths in which the cause is unknown [7] by proportionally distributing these injury deaths to apparent manner of death (homicide, suicide and transport and other unintentional) by age and sex.

We used generalised linear models and linear regression models to assess the association between sex and victim characteristics including age and mechanism of injury (guns, sharp and blunt force) within and between the current survey (2017) and the 2009 study, [2] which

used sampling methods that were comparable at a national level. Coefficients or relative risk (RR) and 95% confidence intervals (CIs) were reported. The models also included interaction terms between gender and year to compare males' and females' homicide characteristics between the two years; p values were reported and associations assessed using a significance level of alpha = 0.05.

### Ethics

Ethical approval for the study was granted by the Ethics Committee of the SAMRC (EC 008-5-2018). Further approval and permission to access data were obtained from the National and Provincial Departments of Health and Forensic Pathology Service.

## Results

A total of 19,477 injury deaths were due to homicide, representing 36% of all injury deaths. Males accounted for 87% of homicides (Table 1). Men had a much higher age standardised homicide rate than women (59.7 vs. 9.0 per 100,000 population), equivalent to 7 male deaths for every 1 female death.

The most common external causes of death were sharp force, firearm discharge and blunt force injuries, with significantly higher rates among men than women (24.4, 20.1 and 11.3 vs 3.1, 2.3 and 1.9/100,000 respectively). Proportionally, more women died due to strangulation or asphyxiation than men (8.9% vs 1.2%), but rates were equivalent.

Men had far higher homicide rates than women in all age groups, specifically among the age group 15–29 (101.2 vs 12.1/100,000 population) and 30–44 (93.0 vs 11.8/100,00 population) years, equating to 8.4 and 7.9 males for every female death in these age groups respectively.

There was considerable interprovincial variation by sex: three to four times higher in the provinces with the highest homicide rates compared to provinces with the lowest rates among both males and females. For males age-standardised homicide rates ranged from 26.7 in Mpumalanga to 100.7/100,000 in the Western Cape; for females the lowest homicide rates were recorded in Limpopo (3.7 per 100,000 population) and the highest in the Eastern Cape (17.5). The male age-standardised homicide rates in the Western Cape were significantly higher than all provinces except the Eastern Cape, which were in turn significantly higher than all other provinces except KwaZulu-Natal. The highest male:female incidence rate ratio (IRR) was recorded in the Western Cape with 11.4 males for every female homicide.

There was also considerable interprovincial variation in the external cause of homicide by sex, particularly for firearm homicides, 88% of which occurred in four provinces (Eastern Cape, KwaZulu Natal, Gauteng and the Western Cape). Sharp force injuries were the leading cause of male homicide in all provinces except Gauteng and the Western Cape, where firearms were the leading cause. For females, sharp force was the leading cause of homicide in all provinces except Mpumalanga.

Temporally, both sexes followed a similar pattern with the highest percentage of cases coinciding with festive periods–December (Christmas) and April (Easter)– and school holidays–July and September. Almost half of the cases were recorded on weekend days compared with weekdays. Disproportionately more men than women were murdered on week and weekend days, particularly on Saturdays (9.3 male deaths for every 1 female death). Rates on Mondays were higher than on other week days. A significantly higher percentage of male homicide victims tested positive for blood alcohol than females (11.4 males for 1 female).

Overall homicide rates could be more than 12% higher than shown in Table 1. Adjusting the age standardised and age specific rates by apportioning additional injury deaths of

**Table 1. Descriptive male and female homicide victim characteristics in South Africa in 2017 by external cause of death, age, province, population group, month of year, day of week and alcohol-relatedness (weighted).**

| | Male | | | M/F Incidence Rate Ratio (95% CI) | Female | | |
|---|---|---|---|---|---|---|---|
| | Number (95%CI) | Percentage (95%CI)** | Age-standardised rate/ 100 000 population (95% CI)* | | Number (95%CI) | Percentage (95%CI)** | Age-standardised rate/ 100 000 population (95% CI)* |
| **All homicides (n = 19477)*** | **16835 (15735, 17936)** | **86.7 (86.2, 87.2)** | **59.7 (55.5, 63.9)** | **6.9 (6.4, 7.4)** | **2583 (2351, 2814)** | **13.3 (12.8, 13.8)** | **9.0 (7.8, 10.1)** |
| External cause (n = 19477)*** | 16835 (15735, 17936) | | | | 2583 (2351, 2814) | | |
| Sharp force | 7071 (6560, 7582) | 42.0 (40.5, 43.5) | 24.4 (22.3, 26.6) | 8.4 (7.5, 9.5) | 885 (771, 999) | 34.3 (32.5, 36.1) | 3.1 (2.4, 3.7) |
| Firearm discharge | 5616 (4955, 6278) | 33.4 (31.3, 35.5) | 20.1 (17.4, 22.9) | 9.0 (7.8, 10.3) | 659 (609, 708) | 25.5 (23.9, 27.2) | 2.3 (1.9, 2.7) |
| Blunt force | 3111 (2940, 3282) | 18.5 (17.0, 20.1) | 11.3 (10.2, 12.4) | 6.0 (5.1, 7.0) | 546 (496, 595) | 21.1 (19.8, 22.6) | 1.9 (1.5, 2.3) |
| Strangled/asphyxiated/ suffocated | 212 (183, 241) | 1.3 (1.1, 1.4) | 0.8 (0.6, 1.1) | 1.0 (0.7, 1.3) | 232 (201, 263) | 9.0 (8.1, 10.0) | 0.8 (0.5, 1.1) |
| Fire /other burn | 125 (90, 160) | 0.7 (0.6, 1.0) | 0.5 (0.2, 0.8) | 3.1 (1.7, 5.5) | 43 (32, 55) | 1.7 (1.2, 2.3) | 0.2 (0.1, 0.2) |
| Other**** | 180 (98, 262) | 1.1 (0.7, 1.6) | 0.6 (0.2, 1.0) | 1.6 (1.1, 2.3) | 122 (77, 167) | 4.7 (3.5, 6.4) | 0.4 (0.3, 0.5) |
| Unknown***** | 520 (435, 606) | 3.1 (2.5, 3.8) | 3.0 (0.8, 5.3) | 5.7 (3.9, 8.2) | 96 (74, 118) | 3.7 (2.9, 4.7) | 0.3 (0.1, 0.6) |
| Age in years (n = 18984) | 16465 (15513, 17416) | | | | 2506 (2304, 2708) | | |
| 0–4 | 153 (102, 204) | 0.9 (0.7, 1.2) | 5.3 (4.4, 6.1) | 1.8 (1.1, 2.8) | 84 (47, 121) | 3.4 (2.3, 4.9) | 3.0 (2.3, 3.6) |
| 5–14 | 122 (101, 143) | 0.7 (0.6, 0.9) | 2.3 (1.9, 2.8) | 1.7 (1.1, 2.8) | 70 (56, 85) | 2.8 (2.3, 3.4) | 1.4 (1.0, 1.7) |
| 15–29 | 7621 (7184, 8058) | 46.3 (45.6, 47.0) | 101.2 (98.9, 103.4) | 8.4 (7.4, 9.4) | 898 (817, 979) | 35.8 (34.4, 37.3) | 12.1 (11.3, 12.9) |
| 30–44 | 6012 (5595, 6429) | 36.5 (35.9, 37.2) | 93.0 (90.6, 95.4) | 7.9 (6.9, 9.0) | 760 (673, 846) | 30.3 (28.9, 32.0) | 11.8 (10.9, 12.6) |
| 45–59 | 1894 (1775, 2012) | 11.5 (10.9, 12.1) | 55.4 (52.9, 57.9) | 5.9 (4.9, 7.0) | 387 (338, 437) | 15.5 (13.6, 17.5) | 9.5 (8.5, 10.4) |
| 60–69 | 461 (417, 505) | 2.8 (2.5, 3.1) | 37.3 (33.9, 40.7) | 3.8 (2.8, 5.2) | 162 (142, 182) | 6.5 (5.6, 7.4) | 9.8 (8.3, 11.3) |
| 70–79 | 163 (137, 190) | 1.0 (0.8, 1.2) | 30.2 (25.6, 34.8) | 3.3 (2.1, 5.2) | 79 (64, 93) | 3.1 (2.7, 3.7) | 9.1 (7.1, 11.1) |
| 80+ | 39 (25, 52) | 0.2 (0.2, 0.3) | 19.1 (13.1, 25.1) | 1.3 (0.7, 2.6) | 66 (47, 84) | 2.6 (1.9, 3.6) | 14.5 (11.0, 18.1) |
| Mean age (SD) | 32.5 (12.7) | | | | 36.4 (17.5) | | |
| Province (n = 19477) | 16835 (15735, 17936) | | | | 2583 (2351, 2814) | | |
| Eastern Cape | 2841 (2667, 3016) | 16.9 (15.5, 18.3) | 97.2 (85.6, 108.8) | 5.4 (4.7, 6.3) | 599 (541, 657) | 23.2 (20.6, 26.0) | 17.5 (12.0, 22.9) |
| Free State | 960 (819, 1100) | 5.7 (4.9, 6.6) | 69.5 (54.9, 84.2) | 7.2 (5.3, 9.7) | 143 (111, 176) | 5.6 (4.4, 7.0) | 10.0 (5.8, 14.2) |
| Gauteng | 3812 (2859, 4764) | 22.6 (18.5, 27.4) | 47.0 (29.9, 64.1) | 6.9 (5.9, 8.0) | 539 (346, 731) | 20.9 (15.5, 27.5) | 7.0 (2.7, 11.4) |

*(Continued)*

**Table 1.** (Continued)

| | Male | | | M/F Incidence Rate Ratio (95% CI) | Female | | |
|---|---|---|---|---|---|---|---|
| | Number (95%CI) | Percentage (95%CI)** | Age-standardised rate/ 100 000 population (95% CI)* | | Number (95%CI) | Percentage (95%CI)** | Age-standardised rate/ 100 000 population (95% CI)* |
| Kwazulu Natal | 3703 (3268, 4138) | 22.0 (19.7, 24.5) | 73.6 (60.9, 86.3) | 6.7 (5.8, 7.7) | 608 (512, 705) | 23.6 (20.3, 27.2) | 10.9 (8.2, 13.7) |
| Limpopo | 640 (465, 815) | 3.8 (2.9, 5.0) | 27.0 (14.8, 39.3) | 6.8 (4.8, 9.7) | 108 (70, 146) | 4.2 (3.1, 5.6) | 3.7 (0.8, 6.6) |
| Mpumalanga | 581 (469, 693) | 3.5 (2.8, 4.2) | 26.7 (18.7, 34.7) | 5.2 (3.7, 7.2) | 118 (91, 145) | 4.6 (3.7, 5.7) | 5.1 (2.5, 7.8) |
| Northern Cape | 211 (107, 314) | 1.3 (0.8, 2.0) | 39.2 (0.0, 80.6) | 4.7 (2.7, 8.1) | 46 (23, 70) | 1.8 (1.1, 3.0) | 8.7 (0.0, 20.0) |
| Northwest | 628 (517, 739) | 3.7 (3.1, 4.5) | 32.2 (19.7, 44.7) | 5.8 (4.1, 8.2) | 105 (102, 108) | 4.1 (3.7, 4.5) | 5.8 (3.4, 8.2) |
| Western Cape | 3460 (3460, 3460) | 20.6 (19.2, 21.9) | 100.7 (100.7, 100.7) | 11.4 (9.4, 13.9) | 316 (316, 316) | 12.2 (11.1, 13.4) | 9.2 (8.8, 9.5) |
| Month of year (n = 19443) | 16808 (15708, 17909) | | | | 2582 (2350, 2813) | | |
| January | 1170 (1104, 1236) | 7.0 (6.7, 7.3) | 4.3 (3.7, 4.8) | 6.7 (5.2, 8.8) | 183 (143, 224) | 7.1 (5.4, 9.3) | 0.6 (0.5, 0.8) |
| February | 1168 (1088, 1248) | 6.9 (6.7, 7.2) | 4.4 (3.9, 4.9) | 6.7 (5.1, 8.7) | 184 (157, 211) | 7.1 (6.2, 8.1) | 0.6 (0.4, 0.9) |
| March | 1378 (1286, 1471) | 8.2 (7.7, 8.7) | 5.1 (4.3, 5.9) | 6.7 (5.3, 8.6) | 215 (193, 237) | 8.3 (7.5, 9.2) | 0.8 (0.5, 1.0) |
| April | 1579 (1480, 1679) | 9.4 (9.0, 9.8) | 5.7 (5.0, 6.4) | 6.3 (5.1, 7.9) | 263 (220, 307) | 10.2 (9.0, 11.5) | 0.9 (0.6, 1.2) |
| May | 1266 (1152, 1380) | 7.5 (7.2, 7.9) | 4.7 (3.9, 5.6) | 5.7 (4.5, 7.2) | 234 (177, 291 | 9.1 (7.6, 10.8) | 0.8 (0.5, 1.1) |
| June | 1205 (1072, 1339) | 7.2 (6.8, 7.6) | 4.4 (3.7, 5.1) | 7.0 (5.4, 9.1) | 181 (135, 227) | 7.0 (5.8, 8.4) | 0.6 (0.4, 0.8) |
| July | 1514 (1359, 1670) | 9.0 (8.6, 9.5) | 5.7 (4.8, 6.7) | 7.0 (5.5, 8.9) | 227 (192, 263) | 8.8 (8.0, 9.6) | 0.8 (0.5, 1.0) |
| August | 1264 (1178, 1351) | 7.5 (7.3, 7.8) | 4.6 (4.0, 5.3) | 7.2 (5.5, 9.3) | 185 (149, 222) | 7.2 (6.2, 8.3) | 0.6 (0.5, 0.8) |
| September | 1519 (1403, 1635) | 9.0 (8.7, 9.4) | 5.6 (4.7, 6.5) | 7.2 (5.7, 9.2) | 221 (201, 241) | 8.6 (7.9, 9.3) | 0.8 (0.5, 1.0) |
| October | 1395 (1310, 1480) | 8.3 (7.9, 8.7) | 5.3 (4.4, 6.2) | 7.2 (5.6, 9.2) | 205 (174, 235) | 7.9 (7.2, 8.7) | 0.7 (0.4, 1.0) |
| November | 1455 (1326, 1585) | 8.7 (8.3, 9.0) | 5.4 (4.6, 6.2) | 7.0 (5.5, 8.9) | 219 (197, 241) | 8.5 (7.7, 9.3) | 0.8 (0.5, 1.0) |
| December | 1894 (1756, 2032) | 11.3 (10.9, 11.7) | 7.0 (6.0, 7.9) | 7.6 (6.1, 9.4) | 264 (239, 289) | 10.2 (9.1, 11.5) | 0.9 (0.7, 1.2) |
| Day of week (n = 19443) | 16808 (15708, 17909) | | | | 2582 (2350, 2813) | | |
| Monday | 2123 (1979, 2267) | 12.6 (12.2, 13.1) | 8.0 (6.9, 9.1) | 5.6 (4.7, 6.8) | 397 (341, 453) | 15.4 (14.2, 16.6) | 1.4 (1.0, 1.7) |
| Tuesday | 1640 (1477, 1803) | 9.8 (9.3, 10.3) | 6.3 (5.2, 7.3) | 5.2 (4.3, 6.4) | 331 (284, 379) | 12.8 (11.8, 13.9) | 1.2 (0.8, 1.5) |
| Wednesday | 1564 (1452, 1676) | 9.3 (8.7, 9.9) | 5.9 (5.0, 6.8) | 5.1 (4.2, 6.3) | 320 (285, 356) | 12.4 (10.9, 14.1) | 1.1 (0.7, 1.5) |

(*Continued*)

**Table 1.** (Continued)

| | Male | | | M/F Incidence Rate Ratio (95% CI) | Female | | |
|---|---|---|---|---|---|---|---|
| | Number (95%CI) | Percentage (95%CI)** | Age-standardised rate/ 100 000 population (95% CI)* | | Number (95%CI) | Percentage (95%CI)** | Age-standardised rate/ 100 000 population (95% CI)* |
| Thursday | 1505 (1378, 1631) | 9.0 (8.7, 9.2) | 5.9 (5.0, 6.8) | 5.7 (4.6, 7.1) | 279 (235, 324) | 10.8 (9.8, 12.0) | 1.0 (0.6, 1.3) |
| Friday | 1918 (1751, 2084) | 11.4 (11.0, 11.8) | 7.2 (6.1, 8.2) | 8.2 (6.5, 10.2) | 247 (217, 277) | 9.6 (8.7, 10.5) | 0.9 (0.6, 1.2) |
| Saturday | 3763 (3508, 4018) | 22.4 (21.7, 23.1) | 13.6 (11.9, 15.2) | 9.3 (7.9, 11.1) | 424 (356, 491) | 16.4 (14.9, 18.1) | 1.5 (1.1, 1.8) |
| Sunday | 4296 (4005, 4588) | 25.6 (24.9, 26.2) | 15.5 (14.0, 16.9) | 7.8 (6.7, 9.0) | 583 (531, 635) | 22.6 (20.5, 24.8) | 2.0 (1.5, 2.5) |
| Blood Alcohol Concentration (n = 3363) ******* | | | | | | | |
| Positive BAC | 1626 (1152, 2099) | 54.8 (53.2, 56.3) | 5.6 (4.2, 7.0) | 11.4 (8.6, 15.2) | 150 (89, 211) | 38.2 (37.1, 39.3) | 0.5 (0.3, 0.7) |
| Mean positive g/100ml (SD; range) | 0.09 (0.10; 0.00, 0.49) | | | | 0.06 (0.08; 0.00, 0.46) | | |

* Column totals do not always sum to *n* due to 59 cases where sex was recorded as 'undetermined'. All rates, except rates for age in years, are age standardised.

** Row percentages are displayed for "All homicide" and column percentages for covariates.

*** Postmortem reports specified a primary cause of death for each death, which we assigned as the external cause. In addition, we noted 2833 cases with multiple injuries, of which 2426 were male, 405 female and 2 unknown.

**** Includes, for males: neglect and abandonment (102 cases), poisoning (45 cases), being pushed from a height (11 cases), drowning/ immersion (9 cases), crushing (2 cases), electrocution (3 cases), and assault by other specified means 8 cases). For females: neglect and abandonment (70 cases), poisoning (21 cases), drowning/ immersion (4 cases), being pushed from a height (2 cases), electrocution (2 cases), maternal deaths/abortion related (5 cases), and assault by other specified means (18 cases).

***** Unknown includes assault by other unspecified means.

****** Blood alcohol results based on the observed data without any adjustment or imputation for missing data (83% of all data)

undetermined intent (Table 2), the overall adjusted homicide rate in South Africa was 7.1 times the global average, and higher among males than females (7.4 vs 5.9 times the global average respectively). There was considerable variation by external cause and age. Notably, IRRs for sharp force injuries amongst males and females were considerably higher than global averages (12.0 and 7.4 times respectively). For males, the highest IRRs were recorded amongst young adults aged 25–34 years, with homicide rates over eight times the global average. For females IRRs were highest in the older age categories, peaking at 10.9 among women older than 70 years.

Male victims were on average four years younger than females in 2009 and 2017 (Table 3).

Males had a significantly higher risk of being killed by sharp force than females in both 2009 and 2017 [RR = 1.46 (9% CI: 1.36, 1.56) vs RR = 1.23 (95% CI: 1.16, 1.46)] respectively. This represented a significant decrease in risk for males and a corresponding increase for females between years. The risk of dying from gunshot injuries increased for both men and women in this period, and was higher for men than women in both 2009 and 2017 [RR = 1.35 (9% CI: 1.27, 1.44) vs RR = 1.31 (95% CI: 1.17, 1.46) respectively. Female risk was higher than men for blunt force injuries [RR = 1.21 (9% CI: 1.12, 1.31) in 2009 vs RR = 1.14 (95% CI: 1.04, 1.24) in 2017], with no change between years. Males had a significantly higher risk of dying on weekends than females in both years [RR = 1.22 (9% CI: 1.16, 1.29) vs RR = 1.23 (95% CI: 1.17,

**Table 2. Male and female homicide rates in South African 2017 mortuary survey (weighted), compared to global rates from the Global Burden of Disease (GBD) study, 2017, [9] by external cause of death and age.**

| | Male | | | | Female | | | | Overall | | | |
|---|---|---|---|---|---|---|---|---|---|---|---|---|
| | Mortuary Survey (MS) | MS Adjusted* (1) | GBD (2) | MS(1): GBD (2) IRR | MS | MS Adjusted* (1) | GBD (2) | MS(1): GBD (2) IRR | MS | MS Adjusted* (1) | GBD (2) | MS(1): GBD (2) IRR |
| | Age-standardised rate/100 000 population (95%CI) | | | Ratio | Age-standardised rate/100 000 population (95%CI) | | | Ratio | Age-standardised rate/100 000 population (95%CI) | | | Ratio |
| All homicide (n = 19477) | 59.7 (55.5, 63.9) | 66.0 (60.8, 71.2) | 8.9 (8.4, 9.5) | 7.4 | 9.0 (7.8, 10.1) | 10.6 (9.1, 11.4) | 1.8 (1.7, 2.0) | 5.9 | 34.0 (31.6, 36.4) | 38.2 (35.2, 41.2) | 5.4 (5.1, 5.7) | 7.1 |
| Firearm discharge | 20.1 (17.4, 22.9) | 22.2 (19.1, 25.5) | 4.2 (4.0, 4.4) | 5.3 | 2.3 (1.9, 2.7) | 2.7 (2.2, 3.1) | 0.5 (0.5, 0.5) | 5.4 | 11.0 (9.6, 12.4) | 12.4 (10.7, 14.0) | 2.3 (2.2, 2.5) | 5.4 |
| Sharp force | 24.4 (22.3, 26.6) | 27.0 (24.4, 29.6) | 2.1 (1.7, 2.3) | 12.9 | 3.1 (2.4, 3.7) | 3.7 (2.8, 4.2) | 0.5 (0.4, 0.5) | 7.4 | 13.9 (12.6, 15.2) | 15.6 (14.0, 17.2) | 1.3 (1.1, 1.4) | 12.0 |
| Other** | 15.1 (13.4, 16.8) | 16.7 (14.7, 18.7) | 2.6 (2.4, 2.9) | 6.4 | 3.6 (3.0, 4.3) | 4.2 (3.5, 4.9) | 0.9 (0.8, 1.0) | 4.7 | 9.3 (8.4, 10.2) | 10.5 (9.4, 11.5) | 1.8 (1.6, 1.9) | 5.8 |
| | Age-specific rate/ 100 000 population (95% CI) | | | Ratio | Age-specific rate/ 100 000 population (95% CI) | | | Ratio | Age-specific rate/ 100 000 population (95% CI) | | | Ratio |
| Age in years (n = 18984) | | | | | | | | | | | | |
| 0–14 | 3.4 (3.0, 3.8) | 4.1 (3.6, 4.6) | 1.5 (1.2, 1.8) | 2.7 | 1.9 (1.6, 2.2) | 2.5 (2.1, 2.9) | 1.1 (0.9, 1.3) | 2.3 | 2.7 (2.4, 2.9) | 3.4 (3.0, 3.6) | 1.3 (1.1, 1.5) | 2.6 |
| 15–19 | 49.8 (46.9, 52.7) | 52.1 (48.7, 55.5) | 10.9 (9.9, 12.0) | 4.8 | 7.0 (5.9, 8.0) | 8.2 (6.8, 9.5) | 1.8 (1.7, 2.1) | 4.6 | 28.2 (26.7, 29.8) | 30.3 (28.5, 32.3) | 6.5 (5.9, 7.1) | 4.7 |
| 20–24 | 120.4 (116.1, 124.7) | 126.7 (121.5, 131.8) | 17.8 (16.7, 19.0) | 7.1 | 12.9 (11.5, 14.3) | 14.7 (12.9, 16.5) | 2.4 (2.2, 2.7) | 6.1 | 66.3 (64.0, 68.5) | 70.7 (67.9, 73.4) | 10.2 (9.6, 10.9) | 6.9 |
| 25–29 | 131.5 (127.1, 135.8) | 139.6 (134.3, 144.8) | 16.3 (15.3, 17.4) | 8.6 | 15.3 (13.8, 16.7) | 17.3 (15.3, 19.1) | 2.3 (2.1, 2.6) | 7.5 | 72.9 (70.7, 75.2) | 78.1 (75.4, 80.9) | 9.4 (8.8, 10.0) | 8.3 |
| 30–34 | 114.1 (109.9, 118.2) | 121.9 (116.8, 126.8) | 15.3 (14.4, 16.3) | 8.0 | 13.0 (11.6, 14.3) | 14.8 (13.0, 16.5) | 2.4 (2.2, 2.6) | 6.2 | 63.1 (60.9, 65.2) | 68.1 (65.4, 70.7) | 8.9 (8.4, 9.5) | 7.7 |
| 35–39 | 88.1 (84.0, 92.1) | 96.7 (91.6, 101.8) | 14.1 (13.3, 15.0) | 6.9 | 11.0 (9.6, 12.5) | 12.6 (10.8, 14.6) | 2.3 (2.1, 2.5) | 5.5 | 49.2 (47.1, 51.4) | 54.4 (51.8, 57.2) | 8.2 (7.8, 8.8) | 6.6 |
| 40–44 | 69.1 (65.3, 73.0) | 74.8 (70.2, 79.6) | 12.0 (11.4, 12.9) | 6.2 | 10.8 (9.2, 12.3) | 12.3 (10.3, 14.3) | 2.2 (2.1, 2.4) | 5.6 | 40.6 (38.4, 42.7) | 44.4 (41.7, 47.0) | 7.2 (6.8, 7.6) | 6.2 |
| 45–49 | 56.9 (53.1, 60.6) | 63.3 (58.5, 68.0) | 10.0 (9.4, 10.6) | 6.3 | 10.9 (9.1, 12.6) | 12.2 (10.0, 14.4) | 2.0 (1.9, 2.1) | 6.1 | 35.4 (33.2, 37.5) | 39.5 (36.7, 42.2) | 6.0 (5.7, 6.4) | 6.6 |
| 50–69 | 35.3 (33.5, 37.1) | 40.6 (38.3, 43.0) | 7.8 (7.4, 8.2) | 5.2 | 12.2 (11.0, 13.4) | 14.4 (12.8, 16.0) | 1.9 (1.8, 2.0) | 7.6 | 25.1 (24.0, 26.3) | 29.1 (27.7, 30.7) | 4.8 (4.6, 5.0) | 6.1 |
| 70+ | 15.4 (13.3, 17.5) | 18.2 (15.4, 21.0) | 5.0 (4.7, 5.4) | 3.6 | 19.4 (16.2, 22.5) | 22.9 (18.8, 27.0) | 2.1 (1.9, 2.2) | 10.9 | 16.8 (15.0, 18.6) | 19.8 (17.5, 22.2) | 3.4 (3.2, 3.6) | 5.8 |
| Mean age (SD) | 32.5 (12.7) | | | | 36 (17.5) | | | | 33.0 (13.6) | | | |

* Adjusted for undetermined cause of unnatural death as presented in Prinsloo et at (2021).

** Includes blunt force, strangled/asphyxiated/suffocated, being pushed from a height, drowning/immersion, poisoning from ingestion, poisoning from gas, fire or other burn, neglect and abandonment, maternal death/ abortion related, crushing, electrocution, assault by other specified means, and unknown cases.

1.29)] respectively, with no change over time. There were no significant differences by sex or year in homicide risk during the hot season.

## Discussion

These findings confirm the enduring nature of South Africa's problem of interpersonal violence in 2017 and the massive, disproportionate homicide risk borne by adult men. This hugely

**Table 3. Comparison of homicide characteristics between 2009 and 2017 by sex and effect measure of study year and sex.**

| Characteristic | Male | | Female | | Effect measure of study year and sex | | |
|---|---|---|---|---|---|---|---|
| | | | | | Sex, RR (95% CI) | | p value |
| | 2009 | 2017 | 2009 | 2017 | 2009 | 2017 | |
| Age, median (IQR) | 29.0 (23.0, 39.0) | 30.0 (24.0, 38.0) | 34.0 (23.0, 47.0) | 32.0 (24.0, 47.0) | Female: 1.00 | Female: 1.00 | |
| | | | | | Male: -4.08 (-4.92, -3.24)* | Male: -3.84 (-4.84, -2.83)* | 0.711 |
| Died from sharp force injuries, percent (95% CI) | 43.8 (41.1, 46.5) | 42.0 (40.5, 43.5) | 30.0 (27.7, 32.5) | 34.2 (32.5, 36.1) | Female: 1.00 | Female: 1.00 | |
| | | | | | Male: 1.46 (1.36, 1.56) | Male: 1.23 (1.16, 1.30) | <0.001 |
| Died from gunshot injuries, percent (95% CI) | 30.1 (27.6, 32.7) | 33.4 (31.3, 35.5) | 22.3 (20.0, 24.9) | 25.5 (23.9, 27.2) | Female: 1.00 | Female: 1.00 | |
| | | | | | Male: 1.35 (1.27, 1.44) | Male: 1.31 (1.17, 1.46) | 0.613 |
| Died from blunt force injuries, percent (95% CI) | 22.1 (20.9, 23.5) | 18.5 (17.0, 20.1) | 26.8 (24.8, 29.0) | 21.1 (19.7, 22.5) | Male: 1.00 | Male: 1.00 | |
| | | | | | Female: 1.21 (1.12, 1.31) | Female: 1.14 (1.04, 1.24) | 0.295 |
| Died on a weekend, percent (95% CI) | 46.0 (44.9, 47.1) | 48.0 (46.8, 49.1) | 37.6 (35.5, 39.7) | 39.0 (37.1, 40.9) | Female: 1.00 | Female: 1.00 | |
| | | | | | Male: 1.22 (1.16, 1.29) | Male: 1.23 (1.17, 1.29) | 0.885 |
| Died during hot season, percent (95% CI) | 42.9 (41.8, 44.0) | 42.0 (41.4, 42.7) | 42.7 (40.6, 44.8) | 41.3 (37.3, 45.4) | Female: 1.00 | Female: 1.00 | |
| | | | | | Male: 1.01 (0.96, 1.05) | Male: 1.02 (0.93, 1.12) | 0.788 |

* coefficient

elevated risk was already reported in previous national estimates in which males accounted for 84% of homicides, and 86% in 2000 and 2009 respectively [2, 10]. Although homicide decreased from 2009 [2]–overall and amongst men and women–the decrease amongst men was proportionally smaller. This is consistent with global data showing that men bear a consistently higher share of homicide than women, [11] but in South Africa the male: female rate ratio is considerably greater. The disaggregated homicide pattern presented in this study is similar to countries in Latin America and the Caribbean with high overall homicide rates (>25 per 100,000 population), largely among men (>80%). Conversely, countries with low homicide rates (<5 per 100,000 population) have a lower proportion (<60%) of male homicides [6]. The fact that men are both perpetrators and victims of homicides masks the strong evidence that men are extremely vulnerable in many contexts. Responding to this inequity impacting men is complicated further by men frequently holding greater power in high violence settings and by targeted public health responses that continue to address violence only on women and children.

Although the risk of homicide was higher for men than women at all ages, age strongly predicted the risk of homicide, with victims being predominantly males between 15–44 years old, and the sex differential starting from a very young age. A previous national survey reported a five times higher homicide rate among boys than girls [12]. The risk factors for interpersonal violence in South Africa are well understood [13, 14]. However, a plethora of co-occurring factors exacerbates the risk of violence in South Africa, as it does in any other high violence setting. These include areas of lower socioeconomic status with greater economic disparities and legacies of colonialism, migrant labour, slavery, other forms of discrimination and human rights violations.

Whereas demographics also determine risk for homicide globally–highest among young adults and males–these risks in South Africa are compounded by socioeconomic factors with high concentrations of homicide in extremely poor neighbourhoods [15]. The country is among the world's most unequal, [16] a legacy of the systemic violence of its post-colonial past, the migrant labour system for mines, and the recent history of racial segregation. Rapid urbanisation has also led to the development of large urban slums that lack the requisite physical and social infrastructure to facilitate social cohesion, with easy access to cheap alcohol [17]. Socially, violence has been normalised as a frequent feature of civil protest and political discourse, and the hegemonic form of masculinity is patriarchal. South Africa also has high levels of legal and illegal firearm ownership and the highest rates of incarceration–an additional exposure to institutional and interpersonal violence–in Africa, with a ratio of male to female prisoners that is double the global average [18]. Given these combination of factors, it is not surprising that this is a society in which interpersonal violence is often expected, and that the forms that it takes are highly gendered. Common reactions to adverse events tend to differ between men and women. Men are socialised into coping by externalising through anger, irritability, violence against intimate partners and others, and increased engagement in risk-taking behaviours [19]. This, alongside the high levels of violence to which males are exposed across the life course, [20] engenders a continuous, and often intergenerational cycle of violence.

South African data have consistently shown that men are not only the main perpetrators of violence, [5] they also have an overwhelmingly higher risk of violent death. Part of this may relate to prevailing gender norms in which men identify with the role of "protector" [21]. Defending honour and asserting dominance over others may increase men's resistance in the face of conflict, in turn increasing the risk of fatal outcomes. This was shown in a recent South African study of co-occurring violence during robbery events in which male victims were significantly more at risk of a fatal outcome [22].

Violence against women is endemic in South Africa, with rates almost six times the global figures. South Africa has responded proactively to such violence with interventions and policy measures culminating in a National Strategic Plan on Gender-based Violence & Femicide, including measures to strengthen the criminal justice system, promote accountability across the state and support survivors. However, men's disproportionate burden of homicide has not resulted in targeted, meaningful prevention. The number of female homicides decreased over time, while the number of male homicides, and hence their share of all homicide, increased from 2009 to 2017 [2]. Yet this has not changed the prevailing socially normative perception that men are neither vulnerable to, nor the victims of, trauma [20]. Ratele et al (2016) suggest that this limited engagement with evidence of men's vulnerability has inadvertently pathologised black males in South Africa, and prevents us from recognising that boys and men are legitimate recipients of violence prevention interventions [23]. There is an urgent need for effective interventions that target men and address not only the gender norms that increase risk, but also the structural drivers of homicide that are rooted in poverty and socio-economic inequality.

In comparison with global averages, South African men, women and children were all exposed to abnormally high levels of homicide risk. Age-standardised homicide rates were similar in 2017 and 2009 [2]. However, the share of overall mortality due to homicide–ranked 8th and accounting for 3.5% of all-cause mortality in 2012 [3]–is set to rise as mortality from other major causes such as HIV continues to decrease, [24] while the share of homicide among all injury deaths is expected to rise due to a decrease in road deaths [7]. The global decline in homicide has also resulted in the relative risk of homicide in South Africa increasing from 5.8 to 7.1 times the global average from 2009 to 2017.

The male to female ratio was highest for firearm discharge, the second leading cause. The implementation of the Firearms Control Act was associated with reductions in firearm homicide, but poor enforcement is associated with a subsequent surge in gun deaths [25]. Males, who are also more likely to be armed, [22] accounted for the larger share of the increase in gun deaths reported between 2009 and 2017. The higher incidence of homicide on weekends and holidays affirmed the prominent role of alcohol, present in 55% of male and 38% of female homicide cases. South Africa's drinking pattern is characterised by very high levels of heavy episodic drinking, particularly amongst males, which is reflected in the gender distribution of homicides attributable to alcohol. Males accounted for 95% of the estimated 15,168 alcohol attributable-homicides in South Africa in 2000, 2006 and 2012 [26]. Alcohol sales bans implemented alongside lockdowns during South Africa's COVID-19 pandemic were associated with reductions in non-natural deaths and trauma cases, [27, 28] but this critical policy window has not, as yet, translated into a more sustained and proactive approach to reduce alcohol harms [29].

There was considerable interprovincial variation in overall homicide. While the homicide risk was higher for men than women in all provinces, the provinces with the highest male homicide rates also ranked highest for female homicide. This suggests violence is endemic in some provinces, which will require complex population-level approaches to prevention that address social determinants and norms that support violence. The temporal pattern, with the highest homicide incidence in months that coincided with school holidays and festive seasons, rather than the warmer months, did not support the theory that aggression (and with it homicide) is associated with increases in temperature [30]. The temperature range from winter to summer in South Africa may be insufficient to affect aggression levels, but further analysis should explore the potential confounding effect of alcohol consumption on the temporal pattern.

South Africa is one of the countries worldwide with a quadruple disease burden where injuries feature alongside major infectious and non-communicable diseases and the largest HIV pandemic worldwide. The burden is compounded by the deleterious effects of violence on other outcomes -mental health and developmental issues such as substance abuse, chronic conditions (e.g. gastrointestinal, gynaecological and fatigue-related), absenteeism and loss of work—and, moreover, is a major impediment to social development [31, 32]. Yet despite its importance among major causes of injury, there is no sense of urgency about addressing violence as a structural issue, and interventions and policies to reduce population-level homicide have been ineffective with the possible exception of intimate femicide, which has the focus of concerted prevention efforts over two decades [33].

With male violence frequently located in the public space, [22] there is potential to converge the prevention agenda to reduce male and female homicide jointly, which might give population-level approaches more impetus. This would also align with the Sustainable Development Goals (SDGs), which have substantially expanded the scope of violence prevention by advocating the reduction of 'all forms of violence everywhere' (Target 16.1), eventually–but only implicitly—recognising men as potential targets for prevention action. The urgent need to address violence against women and children should therefore be integrated into an inclusive approach to address violence in line with SDG16 [34].

There remains an urgent need for homicide and other violence indicators to be disaggregated by sex, to shed further light on socio-demographic-specific risk factors and situational pathways to homicide in various types of violence. This is not only the case in South Africa. The WHO explains the critical importance of disaggregating data for health systems, notably to provide information to allocate appropriate resources; yet the WHO itself only started reporting disaggregated global health statistics in 2019. The forthcoming male homicide study,

the second phase of the current study, will be the first research to profile male victims and perpetrators.

The study has several limitations. The sample size was adequate for estimating population incidence rates in 2009 and 2017 at the national level, but the study lacked power to compare rates for certain subgroups between study years. In addition, the sampling frame was different between the two years, with a smaller sampling frame in 2009, devised to compare injury rates across metropolitan and non-metropolitan populations but limiting our ability to compare provincial homicide patterns. Another limitation was that with only two time points we could not test for trends in male homicide rates. The large number of cases missing blood alcohol data (83% of all cases) requires that these findings be interpreted with caution. Despite these limitations, our study demonstrates the value of mortuary-based survey data for estimating sex-disaggregated homicide data in the absence of routine injury mortality surveillance and confirms that this approach is feasible in a high violence, resource-limited setting.

## Public health implications

Our study highlights the extraordinarily high levels of homicide in South Africa, the disproportionate burden borne by adult men, and the negligible evidence-based prevention response to date. We urgently need a redoubling of efforts to control alcohol and firearms, which have already been shown to influence rates of violence in South Africa, as well as programmes to address the insidious effect of societal norms that drive the excessive burden of physical violence borne by men, and structural interventions to overcome the root causes of poverty and inequality. We consider our study to be an important and necessary–if belated—first step to identify specific groups at increased homicide risk who could benefit from specific interventions and policies. Only through challenging the perceived invulnerability of males can we begin to address the enormous burden of violence borne by men.

## Supporting information

**S1 Table. External cause of homicide categories included in the injury-related mortality survey and corresponding ICD-10 codes, South Africa, 2017.**
(DOCX)

**S1 Text. Inclusivity in global research.**
(DOCX)

## Acknowledgments

We are grateful for the support of forensic pathology services across all provinces and, in particular, to Professor Emeritus Gert Saayman of the Faculty of Health Sciences, Dept of Forensic Medicine, University of Pretoria; Prof Jeanine Vellema, the Retired Head of the Clinical Department of the Gauteng Department of Health Forensic Pathology Service Southern Cluster and an Honorary Member of the University of the Witwatersrand Division of Forensic Medicine & Pathology; and Dr Sibusiso Ntsele of the eThekwini Forensic Pathology Services, KwaZulu-Natal Department of Health, Durban, South Africa.

## Author Contributions

**Conceptualization:** Richard Matzopoulos, Brett Bowman, Rachel Jewkes, Naeemah Abrahams.

**Data curation:** Megan R. Prinsloo, Shibe Mhlongo, Lea Marineau, Nomonde Gwebushe.

**Formal analysis:** Richard Matzopoulos, Megan R. Prinsloo, Shibe Mhlongo, Lea Marineau, Nomonde Gwebushe, Asiphe Ketelo, Carl Lombard.

**Funding acquisition:** Richard Matzopoulos, Bianca Dekel, Rachel Jewkes, Naeemah Abrahams.

**Investigation:** Richard Matzopoulos, Thakadu A. Mamashela, Asiphe Ketelo, Lorna J. Martin.

**Methodology:** Richard Matzopoulos, Megan R. Prinsloo, Shibe Mhlongo, Lorna J. Martin, Bianca Dekel, Carl Lombard, Naeemah Abrahams.

**Project administration:** Megan R. Prinsloo, Asiphe Ketelo.

**Resources:** Richard Matzopoulos, Thakadu A. Mamashela, Lorna J. Martin, Bianca Dekel, Rachel Jewkes.

**Supervision:** Richard Matzopoulos, Naeemah Abrahams.

**Validation:** Naeemah Abrahams.

**Writing – original draft:** Richard Matzopoulos.

**Writing – review & editing:** Richard Matzopoulos, Megan R. Prinsloo, Shibe Mhlongo, Lea Marineau, Morna Cornell, Brett Bowman, Thakadu A. Mamashela, Nomonde Gwebushe, Asiphe Ketelo, Lorna J. Martin, Bianca Dekel, Carl Lombard, Rachel Jewkes, Naeemah Abrahams.

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
