## [Decision Letter · Decision Letter 0]

24 Jul 2023

PGPH-D-23-01105

South Africa’s male homicide epidemic hiding in plain sight: exploring sex differences and patterns in homicide risk in a retrospective descriptive study of postmortem investigations

Dear Dr. Matzopoulos,

Thank you for submitting your manuscript to PLOS Global Public Health. After careful consideration, we feel that it has merit but does not fully meet PLOS Global Public Health’s publication criteria as it currently stands. Therefore, we invite you to submit a revised version of the manuscript that addresses the points raised during the review process.

We look forward to receiving your revised manuscript.

Kind regards,

Alok Atreya

Academic Editor

Journal Requirements:

2. Please send a completed 'Competing Interests' statement, including any COIs declared by your co-authors. If you have no competing interests to declare, please state "The authors have declared that no competing interests exist". Otherwise please declare all competing interests beginning with twhe statement "I have read the journal's policy and the authors of this manuscript have the following competing interests:"

3. We noticed that you used "data not shown" in the manuscript. We do not allow these references, as the PLOS data access policy requires that all data be either published with the manuscript or made available in a publicly accessible database. Please amend the supplementary material to include the referenced data or remove the references.

4. We do not publish any copyright or trademark symbols that usually accompany proprietary names, eg  ©, ®, ™  (e.g. next to drug or reagent names). Please remove all instances of trademark/copyright symbols throughout the text, including © on page 11.

Additional Editor Comments (if provided):

Overall, this is a well-written manuscript presenting important data on the disproportionate burden of homicide among males in South Africa. The methods are sound and appropriately described, the results are clearly presented, and the discussion provides good context for interpreting the findings. The authors make a compelling case that more attention needs to be paid to violence prevention among men and boys in South Africa. I have only a few minor suggestions:

Methods:

- The sampling methods and case selection criteria are clearly described. However, it would be helpful to state the final sample size included in the analyses after exclusions.

- Were any steps taken to address missing data? This is not discussed but seems relevant for variables like blood alcohol concentration.

Results:

- The results are comprehensive and informative. The tables effectively summarize key findings.

- It would be interesting to see age-stratified rates for different mechanisms of homicide (e.g. firearm injuries) between males and females. Are the differences most pronounced in certain age groups?

Discussion:

- The discussion contextualizes the results well and makes a strong case for the need to address violence against men and boys.

- When comparing to the previous 2009 national study, it would be helpful to comment on whether the sampling methods were comparable between the two time points.

- Were statistical tests done to assess changes over time? If so, it would be useful to report whether observed differences were statistically significant.

Overall this is a nicely written paper that makes an important contribution to the literature on violence and injuries in South Africa. My suggestions are minor and intended to provide a bit more methodological detail.

Reviewers' comments:

Reviewer's Responses to Questions

**Comments to the Author**

1. Does this manuscript meet PLOS Global Public Health’s publication criteria? Is the manuscript technically sound, and do the data support the conclusions? The manuscript must describe methodologically and ethically rigorous research with conclusions that are appropriately drawn based on the data presented.

Reviewer #1: Yes

Reviewer #2: Yes

Reviewer #3: No

2. Has the statistical analysis been performed appropriately and rigorously?

Reviewer #1: Yes

Reviewer #2: Yes

Reviewer #3: Yes

3. Have the authors made all data underlying the findings in their manuscript fully available (please refer to the Data Availability Statement at the start of the manuscript PDF file)?

Reviewer #1: Yes

Reviewer #2: Yes

Reviewer #3: No

4. Is the manuscript presented in an intelligible fashion and written in standard English?

Reviewer #1: Yes

Reviewer #2: Yes

Reviewer #3: Yes

5. Review Comments to the Author

Reviewer #1: 1. why is data from only 8 out of 9 provinces were collected?

2. briefly discuss about alcohol test. what was teh test, how, when because there are likelihood of error and bias.. since it is mentioned as (1) objective in the introduction section.

Reviewer #2: The authors have tackled a very important public health issue in South Africa. In addition, they have conducted a very well-designed study and have presented the findings excellently. They deserve our congratulations. I do not have any methodological critiques of their paper – only minor comments below:

1. What proportion of homicides were due to multiple mechanisms (blunt and penetrating for example)?

2. There are a few typos on page 5 – “homides” instead of “homicides”; missing word in the second sentence in the Discussion section. Another round of copyedits will suffice.

Congratulations once again.

Reviewer #3: Thank you for submitting this original study focused on understanding the burden of homicide in the South African context. This study retrospectively analyzed data from postmortem investigations, and focused on sex specific differences in rates of homicide. The conclusions of the study focus on the fact that men are significantly more likely to be victims of homicide than females, and the statistical methods focus on this as the main conclusions.

While the data represents a very interesting source of information, the conclusions need to be considerably refocused. The comparison between homicide in men and women while justified in theory by the data, does not really provide any helpful information that should educate future interventions. The focus on gender is a missed opportunity in terms of use of this data, and this paper would benefit from being focused more on the high rates of homicide in men and specific sources of homicide rooted in geography, sociodemographic characteristics other than gender, and a more data-driven conclusion. The authors seem to interject a lot of opinion in the study regarding poverty, inequality, firearm use, and alcohol use, but this is not really discussed in full detail in the study because of the focus on gender.

1. Focus the study on root causes of homicide in men, and remove the focus on comparisons between men and women. An unintended consequence of this study would be to defund initiatives focused on homicide and injury prevention in women and children, so this manuscript needs to be completely rewritten to prevent this sort of downstream policy consequence.

2. The conclusions need to be much more data driven. This could include a focus on geographic differences and associated differences in policy, firearm usage and policy changes over time within each of the regions affected, and other such detailed analyses of the data that can be more helpful from a policy standpoint.

3. The authors must confirm that the data is publicly available based on the guidelines of the PLOS journals.

6. PLOS authors have the option to publish the peer review history of their article (what does this mean?). If published, this will include your full peer review and any attached files.

**Do you want your identity to be public for this peer review?** For information about this choice, including consent withdrawal, please see our Privacy Policy.

Reviewer #1: No

Reviewer #2: No

Reviewer #3: No

---

## [Decision Letter · Decision Letter 1]

18 Oct 2023

South Africa’s male homicide epidemic hiding in plain sight: exploring sex differences and patterns in homicide risk in a retrospective descriptive study of postmortem investigations

PGPH-D-23-01105R1

Dear Dr Matzopoulos,

We are pleased to inform you that your manuscript 'South Africa’s male homicide epidemic hiding in plain sight: exploring sex differences and patterns in homicide risk in a retrospective descriptive study of postmortem investigations' has been provisionally accepted for publication in PLOS Global Public Health.

Best regards,

Alok Atreya

Academic Editor

Reviewer Comments (if any, and for reference):

Reviewer's Responses to Questions

**Comments to the Author**

1. If the authors have adequately addressed your comments raised in a previous round of review and you feel that this manuscript is now acceptable for publication, you may indicate that here to bypass the “Comments to the Author” section, enter your conflict of interest statement in the “Confidential to Editor” section, and submit your "Accept" recommendation.

Reviewer #1: All comments have been addressed

Reviewer #2: All comments have been addressed

2. Does this manuscript meet PLOS Global Public Health’s publication criteria? Is the manuscript technically sound, and do the data support the conclusions? The manuscript must describe methodologically and ethically rigorous research with conclusions that are appropriately drawn based on the data presented.

Reviewer #1: Yes

Reviewer #2: Yes

3. Has the statistical analysis been performed appropriately and rigorously?

Reviewer #1: Yes

Reviewer #2: Yes

4. Have the authors made all data underlying the findings in their manuscript fully available (please refer to the Data Availability Statement at the start of the manuscript PDF file)?

Reviewer #1: Yes

Reviewer #2: Yes

5. Is the manuscript presented in an intelligible fashion and written in standard English?

Reviewer #1: Yes

Reviewer #2: Yes

6. Review Comments to the Author

Reviewer #1: I am satisfied with all the answers except the alcohol concentration. Only the test has been mentioned. It would have made a better impact if it were to include the method of collection and inclusion and exclusion criteria for it because where it can present as an artefact.

Reviewer #2: (No Response)

7. PLOS authors have the option to publish the peer review history of their article (what does this mean?). If published, this will include your full peer review and any attached files.

**Do you want your identity to be public for this peer review?** For information about this choice, including consent withdrawal, please see our Privacy Policy.

Reviewer #1: No

Reviewer #2: No
